

# Joint intent detection and slot filling with syntactic and semantic features using multichannel CNN-BiLSTM

Yusuf Idris Muhammad, Naomie Salim and Anazida Zainal

Faculty of Computing, Universiti Teknologi Malaysia, Skudai, Johor, Malaysia

## ABSTRACT

Understanding spoken language is crucial for conversational agents, with intent detection and slot filling being the primary tasks in natural language understanding (NLU). Enhancing the NLU tasks can lead to an accurate and efficient virtual assistant thereby reducing the need for human intervention and expanding their applicability in other domains. Traditionally, these tasks have been addressed individually, but recent studies have highlighted their interconnection, suggesting better results when solved together. Recent advances in natural language processing have shown that pretrained word embeddings can enhance text representation and improve the generalization capabilities of models. However, the challenge of poor generalization in joint learning models for intent detection and slot filling remains due to limited annotated datasets. Additionally, traditional models face difficulties in capturing both the semantic and syntactic nuances of language, which are vital for accurate intent detection and slot filling. This study proposes a hybridized text representation method using a multi-channel convolutional neural network with three embedding channels: non-contextual embeddings for semantic information, part-of-speech (POS) tag embeddings for syntactic features, and contextual embeddings for deeper contextual understanding. Specifically, we utilized word2vec for non-contextual embeddings, one-hot vectors for POS tags, and bidirectional encoder representations from transformers (BERT) for contextual embeddings. These embeddings are processed through a convolutional layer and a shared bidirectional long short-term memory (BiLSTM) network, followed by two softmax functions for intent detection and slot filling. Experiments on the air travel information system (ATIS) and SNIPS datasets demonstrated that our model significantly outperformed the baseline models, achieving an intent accuracy of 97.90% and slot filling F1-score of 98.86% on the ATIS dataset, and an intent accuracy of 98.88% and slot filling F1-score of 97.07% on the SNIPS dataset. These results highlight the effectiveness of our proposed approach in advancing dialogue systems, and paving the way for more accurate and efficient natural language understanding in real-world applications.

## INTRODUCTION

Advancements in artificial intelligence have led to the development of intelligent agents known as dialogue systems that can engage in conversations with humans and assist in

Corresponding author
Yusuf Idris Muhammad,
muhammadidris@graduate.utm.my

daily tasks. A core component of these systems is natural language understanding (NLU), which enables interactions between human beings and dialogue systems (*Weld et al., 2022*). NLU aims to extract meaning from a user's utterances and to infer their intentions. The two primary tasks in NLU are intent detection and slot filling. An intent detection is a classification task, whereas slot filling is a sequence labeling task. Traditionally, these tasks have been handled separately, but it has been shown that modeling them together yields better performance (*Suhaili, Salim & Jambli, 2021*). This joint approach captures both intent and slot label distributions within an utterance considering both local and global contexts (*Louvan & Magnini, 2020*). Unlike separate models, the joint model reduces error propagation by using a single model for training and fine-tuning, which can enhance intent detection and slot filling performance (*Firdaus, Ekbal & Cambria, 2023*).

Improvements in intent detection and slot filling performance have significant importance and potential applications in real-world scenarios. For instance, in customer service, enhanced NLU models can lead to accurate and efficient virtual assistants, thereby reducing the need for human intervention and improving customer satisfaction (*Huang & Rust, 2021*). Moreover, advancements in intent detection and slot filling not only enhance the performance of dialogue systems but also expand their applicability and effectiveness in various domains, leading to more intuitive and efficient human-machine interaction (*Hao et al., 2023*; *Lim et al., 2022*; *Wang et al., 2023*; *Zhou et al., 2022*).

Encoder–decoder neural network architectures are often used for joint learning classification owing to their strong sequential processing capabilities. However, these models face challenges such as generalization issues and the need for substantial annotated datasets. To address generalization, transfer learning with pretrained language models is utilized, starting with a model pretrained on a large corpus and then fine-tuning it on a specific task with domain-specific data. Most existing methods for joint models use either non-contextual embeddings (*Bhasin et al., 2020*; *Firdaus et al., 2021*; *Pan et al., 2018*) or contextual embeddings (*Ni et al., 2020*; *Qin et al., 2019*) to address these issues. Non-contextual embeddings provide foundational representations that are useful for initializing embedding layers, capturing word-level semantics, reducing dimensionality, and handling out-of-vocabulary words. However, these embeddings can mislead the model's predictions owing to the distributional hypothesis, which assumes that words appearing in similar contexts have similar meanings. For instance, in the latent space "bad" and "good" are mapped close together as neighbors, which can be problematic for intent detection where semantics and context are crucial. In addition, non-contextual embeddings struggle to differentiate between the different meanings of polysemous words, leading to ambiguity in intent detection and slot filling tasks. For instance, in the utterance "Add Brian May to my reggae infusion list," the model might incorrectly label the slot for "May" as "timeRange" which negatively impacts the performance of the model. On the other hand, contextual embeddings capture the meaning of words within the context of the surrounding words in a sentence, offering better adaptability for specific tasks. However, they are limited by the vocabulary used during pretraining. To enhance the understanding of the model, some studies have incorporated syntactic representations of utterances alongside word vectors (*Guo et al., 2014*). For example, part-of-speech (POS) tag embeddings encode syntactic

properties and provide grammatical roles, and help the model disambiguate words with multiple meanings (*Wang et al., 2021*). Therefore, we assert that combining contextual, non-contextual, and syntactic features is essential for effective feature extraction in joint learning classification for intent detection and slot filling. This study explores the impact of these embeddings on the joint model and compared the impact of domain-specific and general embeddings.

Earlier approaches to joint models relied on statistical methods that could not capture deep semantic information. Deep neural networks have revolutionized natural language processing tasks such as intent detection and slot filling. Common models for intent detection include convolutional neural networks (CNNs), recurrent neural networks (RNNs), and transformer models such as bidirectional encoder representations from transformers (*Khattak et al., 2021*). For the slot filling task, RNNs and encoder–decoder models have shown promising results (*Liu & Lane, 2016a*; *Siddique, Jamour & Hristidis, 2021*). While RNNs capture the chronological features of utterances, they often miss the local semantic features that are crucial for accurate slot identification. CNNs, on the other hand, effectively capture local semantic information and generate higher-level representations (*Kim, 2014*), but they do not capture sequential dependencies, which are essential for analyzing natural language.

Motivated by these issues, this study proposes a joint learning classification model that utilizes a multichannel convolutional neural network (MCNN) to extract features from three embedding layers: contextual (BERT), non-contextual (word2vec), and POS tag embeddings. These features were encoded using a shared BiLSTM to retain sequential correlations and capture the global dependencies in the utterances. The output from the BiLSTM is then fed into the intent and slot filling decoders for intent detection and slot filling. The major contributions of the proposed model are summarized as follows:

1. We introduced a joint model that employs an MCNN with three input channels: contextual, non-contextual, and POS tag embeddings. This setup extracts contextual, semantic, and syntactic information for intent detection and slot filling tasks.
2. We examined the significance of different embeddings in the context of joint learning classification.
3. We analyzed the impact of using general embeddings *versus* domain-specific embeddings on the air travel information system (ATIS) and SNIPS datasets.
4. We conducted a series of experiments on the ATIS and SNIPS datasets and demonstrated that our approach outperformed the baseline methods.

The remainder of this paper is organized as follows: In sections, "Related Work," "Proposed model," "Experimental Study," "Experimental Results and Analysis" and "Conclusion and Future Work."

## RELATED WORK

### Classical approaches

Joint learning classification for intent detection and slot filling has been widely studied by many researchers. *Jeong & Lee (2008)* conducted one of the earliest studies on this topic

(*Jeong & Lee, 2008*). They used a triangular-chain conditional random field (CRF) to model the dependencies between slots and intent. In another study conducted by *Wang (2010)*, a maximum entropy model was used for intent detection and CRF for slot-filling tasks. In addition, (*Celikyilmaz & Hakkani-Tur, 2012*) presented a multilayer hidden Markov model for joint learning tasks. Although classical models can successfully capture the dependencies between slots and intent, they face scalability challenges, particularly when dealing with large datasets (*Cohn, 2007*; *Jia, Liang & Liang, 2023*; *Mairesse et al., 2009*). Classical models often involve mathematical computation during training and prediction. As the size of the dataset increases, these computations become computationally expensive, making it difficult to scale models efficiently. They also encountered issues with feature creation, as features needed to be predefined by humans before model training could occur (*Weld et al., 2022*; *Xu & Sarikaya, 2013*; *Zhang & Wang, 2016*). The manual feature engineering process is time-consuming and limits the ability of these models to adapt to the complexity of natural language (*Ferrario & Naegelin, 2020*). These limitations have led to the development of alternative approaches, particularly deep-learning approaches.

## Deep learning approaches

The first study to exploit deep neural networks in a joint learning model for intent detection and slot filling was conducted by *Xu & Sarikaya (2013)*. They utilized a CNN for feature extraction and a CRF for analysis. *Guo et al. (2014)* employed recursive neural networks to analyze the dependency structures of utterances, incorporating features such as n-grams and named entities, and used tree-derived features for intent detection and slot filling. Although their model achieved state-of-the-art performance, it lacks bidirectional connections, leading to slow and ambiguous models. Many subsequent studies have employed different methodologies using RNNs for joint learning classification owing to their suitability for capturing temporal dependencies. In some studies, RNNs were used as word-level classifiers, with intermediate hidden states used for slot filling tasks and the weighted sum of the hidden states for intent detection. In other studies, RNNs functioned as sentence classifiers, using the last hidden state was used for intent detection. *Zhou et al. (2016)* proposed a hierarchical LSTM (HLSTM) with two layers: the bottom-layer served as a sentence classifier, with the last hidden state was employed for intent detection, while the upper layer performed word-level classification for slot labeling. *Liu & Lane (2016b)* introduced a joint model that using LSTM cell. In their proposed model, the intent and slots are detected at each time step as the input arrived, and the overall intent was obtained using the last hidden state. *Hakkani-Tür et al. (2016)* tagged each utterance with a special token before passing it to a BiLSTM to obtain a latent semantic representation of the entire input utterance for intent detection along with intermediate hidden states for slot labeling. *Liu & Lane (2016a)* proposed another joint model that used a context vector generated from the weighted sum of RNN hidden states to obtain both intent and slots. *Zhang & Wang (2016)* combined a gated recurrent unit and max pooling layer in their joint model. *Chao, Ke & Xiaofei (2020)* employed a BiLSTM model with POS tag as embeddings, while *Firdaus et al. (2018)* used a bidirectional gated recurrent unit model with hybridized non-contextual

and POS tag embeddings. Most recently, *Hardalov, Koychev & Nakov (2023)* proposed a transformer based model incorporating named entity recognition as a feature.

## Hybrid models

Our approach is closely related to the work in *Firdaus et al. (2019)*, where two non-contextual embeddings were used. Glove and word2vec were concatenated and served as features for a CNN to capture the sentence representation. This was then fed into a BiLSTM to obtain contextual information within utterances. However, this method's reliance on non-contextual embeddings is a notable drawback. In another study conducted by *Qin et al. (2021)*, a hybrid model combining BERT and BiLSTM was proposed to establish bidirectionality between intent detection and slot filling tasks. However, this study considered only contextual embeddings.

In nutshell, while classical models laid the foundation for intent detection and slot filling, their scalability and manual feature engineering requirements limit their applicability to large datasets and complex language tasks. Deep learning approaches, particularly those leveraging RNNs and CNN, have significantly advanced the field by automating feature extraction and capturing temporal dependencies. However, these models still face challenges such as limited contextual understanding and ambiguity in bidirectional connections. Hybrid models have attempted to bridge these gaps by combining various types of embeddings and model architectures. Nevertheless, existing hybrid models often rely on non-contextual embeddings, which fail to fully exploit the contextual nuances of languages. Moreover, the integration of syntactical features such as POS tags remains underexplored. Our proposed model addresses these issues by integrating contextual, non-contextual, and syntactic embeddings using a multichannel CNN architecture. This approach not only enhances feature representation but also disambiguates polysemous words, leading to improved performance in intent detection and slot filling tasks. To the best of our knowledge, this is the first study to use an MCNN architecture that leverages contextual, non-contextual, and POS tag embeddings in different channels to improve the performance of joint learning classification for intent detection and slot filling.

## Proposed model

Figure 1 illustrates the architecture of the proposed joint model for intent detection and slot filling. This model integrates both the CNN and BiLSTM, as used in *Wu et al. (2024)*. The proposed model comprises three input layers, three convolutional layers with subsequent max pooling, a shared BiLSTM encoder, and two dense layers that implement softmax functions. These components jointly detect the intent of the user's input utterance and the associated slots by assigning them with multiclass labels (B, I, O), where "I," "O," and "B" signify Inside, Outside, and Beginning of slots, respectively. Details of the model are described in the following subsections.

## Input layer (embedding layer)

The proposed model utilizes three types of embeddings: non-contextual, contextual, and POS tag embeddings each as an independent input channel for the network. This layer converts each word in the input sentence into its corresponding vector.

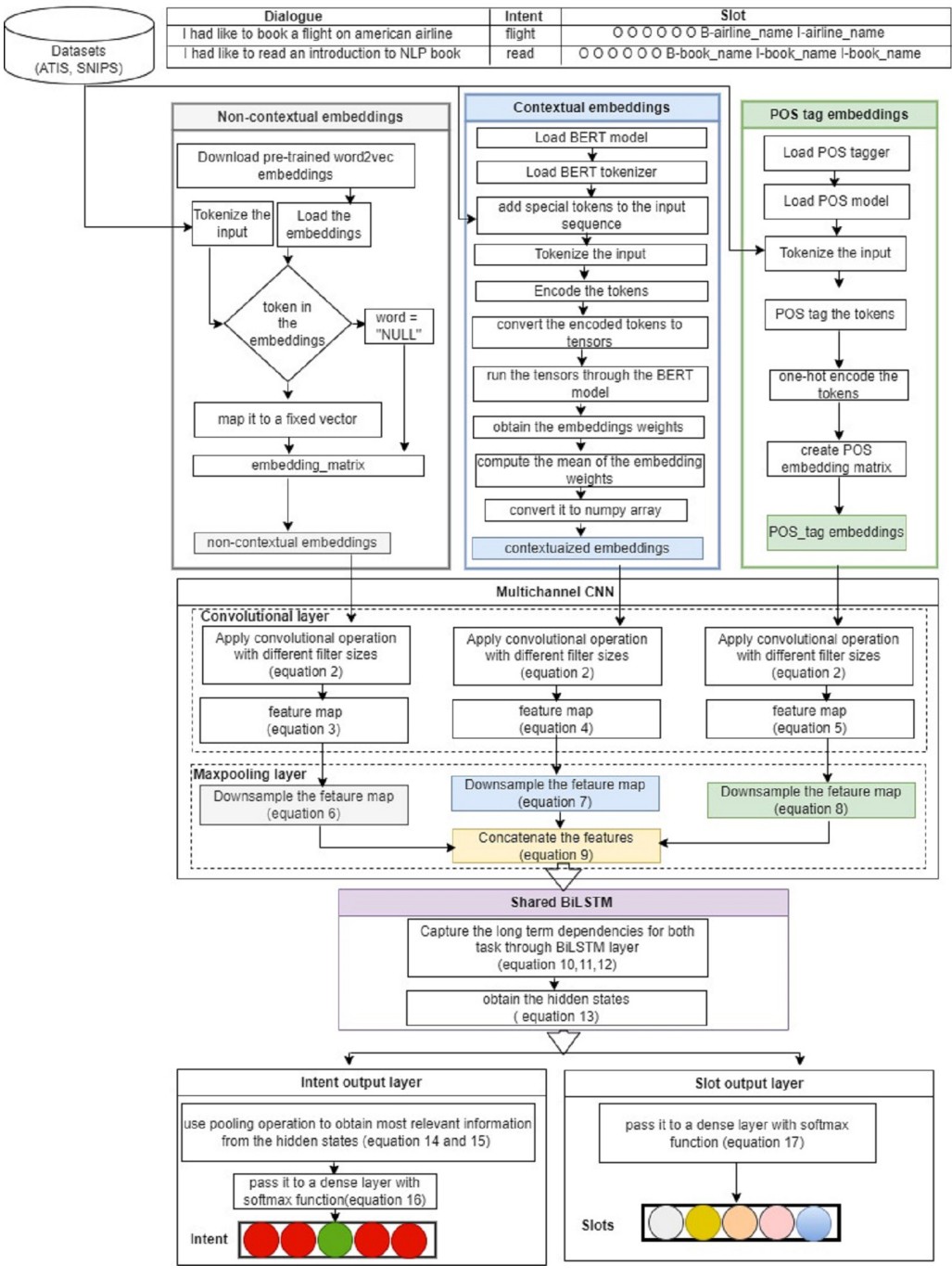

**Figure 1** MCNN-BiLSTM joint model architecture for intent detection and slot filling.

### Non-contextual embeddings channel

The main purpose of this channel is to capture the semantic information of input words. In this study, we used two types of non-contextual embeddings: general pretrained embeddings and domain-specific embedding. For the general embeddings, we used pretrained word2vec

embeddings trained on 100 billion words from Google News (*Mikolov, Yih & Zweig, 2013*), whereas for domain-specific embedding, we used the continuous bag of words architecture trained on the ATIS and SNIPS datasets. Each word is represented by 300-dimensional vectors, with word padding applied to ensure uniform sentence length. Formally, to capture the semantic features of each word for an utterance of length $n$, $i$th word is mapped to a $d$-dimensional embedding expressed as:

$$X_1^n = \{x_1, x_2, \ldots, x_n\} \, for \, X \in \mathbb{R}^d \tag{1}$$

### Contextual embeddings channel

The idea behind using this channel is to capture the contextual representations of the input sentences, which is crucial for intent detection tasks (*Roma et al., 2023*). For this purpose, we used a pretrained bert-base-uncased model developed by Google and pretrained on Wikipedia and BookCorpus corpora (*Devlin et al., 2018*). The model had 12 stacked layers, 768 hidden units, 12 attention heads, and 110 million parameters. We prepared our data by adding special tokens, [CLS] and [SEP], at the beginning and end of each sentence, respectively, to denote the start and end of each sequence. Each sentence was then tokenized to obtain the corresponding tokens for each word, which were then encoded into their corresponding numerical identifiers based on the BERT vocabulary. These identifiers are then converted into tensors and fed into the pretrained BERT model to obtain the embedding weights for each token through the hidden states of the last layer. However, in this study, we computed the means of the hidden states to obtain a fixed size representation of the entire sequence, which served as our contextualized embeddings.

### POS tag embeddings channel

The main purpose of this channel is to incorporate syntactic representations of sentences alongside word vectors to help our model disambiguate words with multiple meanings. In this study, we employed a tokenization function from the natural language tool kit (NLTK) library to split each sentence into its words. Each word is then assigned a POS tag using a POS tagger from the library, which indicates the syntactic role of the word in the sentence. These POS tags were then converted into one-hot encoded vectors. These vectors were then aggregated to create a POS tag embedding matrix for each sentence. This matrix represents the syntactic structures of the sentence, where each row corresponds to a word, and its respective POS tag. Formally, the sequence of POS tags can be represented as:

$$|S|_n^1 = \{s_1, s_2, \ldots, s_n\}, \, with \, s \in R^{36}$$ where 36 denotes the dimensionality of the vector.

## Convolutional layer

After encoding all the sentences into vectors and applying zero padding to ensure a uniform length across all embedding channels. Convolution operations are then applied to extract local features from the embedding channels. This allows the model to identify and leverage important patterns and dependencies in the data. In this study, we applied convolution operations with different filter sizes to POS embeddings ($P$), contextual embeddings ($Q$),

and non-contextual embeddings ($R$). Conventionally, convolution operations involve sliding filters over embeddings to detect patterns and to generate feature maps.

Let $w_n \in R^{hws}, w_c \in R^{hws}, w_p \in R^{hws}$ be filters applied for the non-contextual, contextual, and POS embedding, respectively, where $ws$ is the window size $h$ is the embedding dimension for each word. Thus, the generated features can be expressed as:

$$Z_i = f(w.x_{i:i+ws-1} + b) \tag{2}$$

where ( $\cdot$ ) is the convolution operator, $f$ is a non-linear activation function (*ReLU* in this case), $b$ is the bias. This function is applied to each window $[x_{1:ws}, x_{2:ws+1}, \ldots, x_{n-ws:n}]$ in each channel. Thus, the feature map for each embedding type is given as:

$$\text{POS features: } z_p = \left[z_1^p, z_2^p, \ldots, z_{n-ws+1}^p\right] \tag{3}$$

$$\text{Contextual features: } z_q = \left[z_1^q, z_2^q, \ldots, z_{n-ws+1}^q\right] \tag{4}$$

$$\text{Non-contextual features: } z_r = \left[z_1^r, z_2^r, \ldots, z_{n-ws+1}^r\right] \tag{5}$$

However, it is important to note that various filters can be employed to capture diverse features from embedding matrices (*Rakhmatulin et al., 2024*).

## Max pooling layer

The pooling operation is designed to reduce the resolution of the feature maps by applying a pooling function to several units within a defined local region, known as the pooling size. This process helps to generalize the features obtained from the convolutional layer (*Zhao & Zhang, 2024*). Specifically, the purpose of using the max pooling layer in this study was to capture the most significant features from the convolutional layer by selecting the maximum value from each feature map segment (*Kim, 2014*). Thus, the maximum values for POS and contextual and non-contextual features are expressed as follows:

$$\tilde{Z}_p = max\left[z_1^p, z_2^p, \ldots, z_{n-ws+1}^p\right] \tag{6}$$

$$\tilde{Z}_q = max\left[z_1^q, z_2^q, \ldots, z_{n-ws+1}^q\right] \tag{7}$$

$$\tilde{Z}_r = max\left[z_1^r, z_2^r, \ldots, z_{n-ws+1}^r\right] \tag{8}$$

After pooling the maximum features for each channel, we obtain the final maximum feature by concatenating the three features using the following equation:

$$Z = \tilde{Z}_P^1 \oplus \cdots \oplus \tilde{Z}_P^m \oplus \tilde{Z}_q^1 \oplus \cdots \oplus \tilde{Z}_q^n \oplus \tilde{Z}_r^1 \oplus \cdots \oplus \tilde{Z}_r^o \tag{9}$$

where $\oplus$ is the concatenator operator and **$m$, $n$, $o$** are the filters for POS tag, contextual, and semantic features respectively.
## Shared BiLSTM layer

Bidirectional long short-term memory network (BiLSTM) is a type of recurrent neural network (RNN) architecture known for its ability to capture long-term dependencies in sequential data. BiLSTM is an extension of the basic LSTM that integrates forward and backward LSTM structures to capture deeper contextual information (*Qi et al., 2024*). Essentially, BiLSTM employs two separate LSTM networks to analyze input sequences. The forward LSTM processes the sequence starting from the first token, whereas the backward LSTM processes it from the final token. Subsequently, the outputs of both LSTM networks are merged at each time step. This arrangement ensures that the output at any given time step encompasses both the preceding and succeeding context from the input sequence, thereby facilitating a more effective capture of long-term dependencies within the sequence. The essence of using BiLSTM in this study is to capture sequential dependencies, which are essential for analyzing natural languages (*Pogiatzis & Samakovitis, 2020*). Thus, the concatenated output of the convolutional layer $Z$ is passed through the BiLSTM layer to generate the output $h_t$.

$$\vec{h}_t = LSTM\left(z_t, \vec{h}_{t-1}\right) \tag{10}$$

$$\overleftarrow{h}_t = LSTM\left(z_t, \overleftarrow{h}_{t-1}\right) \tag{11}$$

$$h_t = \vec{W}_t.\vec{h}_t + \overleftarrow{W}_t.\overleftarrow{h}_t + b \tag{12}$$

The output of the BiLSTM is given as:

$$H = [h_1, h_2, \ldots, h_n] \tag{13}$$

## Output layer
### Intent detection output

The intent output is computed using a max pooling layer to acquire the most relevant representation of the entire sequence from BiLSTM, as used by *Zhang & Wang (2016)*. However, in this study, the output of the max pooling layer was passed to a global max pooling layer to provide a global summary of the entire sequence. A fully connected layer with softmax activation was used to detect the intent. Therefore, the intent output vector is computed as follows:

$$h_{maxpool} = \{\max(h_1, h_2), \ldots, \max(h_{n-1}, h_n)\} \tag{14}$$

$$h_{global} = \max(h_1, h_2, \ldots, h_n) \tag{15}$$

$$y^i = Softmax\left(W.h_{global} + b\right) \tag{16}$$

where $h_{maxpool}$, is the max pooled hidden state, $h_{global}$ is the global summary of the sequence, $y^i$ is the intent label, $W$ is the transformation matrix, and $b$ is the bias vector.

### Slot filling output

To obtain the slots, we passed the output of the BiLSTM to a fully connected layer that used softmax as an activation function and computed the output vector as follows:

$$y_i^s = softmax\left(W.h_i + b\right) \tag{17}$$

where $y^s$ is the slot label and $W, b$ are the transformation matrix and bias vectors, respectively.

## Model versions

Our proposed model for the joint learning classification of intent detection and slot filling encompasses seven versions, each of which is tailored to different settings of embeddings passed to different channels in the MCNN model. These versions are described below.

- MCNN-BiLSTM-1: This version primarily aims to assess the influence of pretrained word embeddings. The non-contextual word embedding channel was initialized randomly, whereas the other channels were ignored. This means that only randomized word embeddings were considered for training.
- MCNN-BiLSTM-2a: In this setting, the non-contextual embedding channel is initialized with pretrained word2vec embeddings with other ignored channels.
- MCNN-BiLSTM-2b: In this setting, the non-contextual embedding channel is initialized with domain-specific word2vec embeddings with other ignored channels.
- MCNN-BiLSTM-3: In this case, only the channel for the BERT contextual embeddings was initialized, and all others were ignored.
- MCNN-BiLSTM-4: Two channels were used for training. Specifically, the contextual embedding channel was initialized using BERT embeddings and the non-contextual embedding channel was initialized using word2vec embeddings.
- MCNN-BiLSTM-5: Two channels were used for training. The non-contextual embedding channel was initialized with word2vec embeddings, and the POS embedding channel was initialized with POS tag embeddings.
- MCNN-BiLSTM-6: In this version, contextual embedding and POS tag embedding channels are initialized using BERT and POS tag embedding, respectively.
- MCNN-BiLSTM-7a: In this version, all three channels are used for training. Specifically, we used domain-specific word2vec embeddings trained on the ATIS and SNIPS datasets as inputs for non-contextual embeddings. This study aimed to assess the impact of general embeddings compared to domain-specific embeddings.
- MCNN-BiLSTM-7b: This variant is similar to that of MCNN-BiLSTM-7a. However, in this case, instead of domain-specific word2vec embedding, we used a general purpose pretrained word2vec embedding as an input to the non-contextual embedding channel.

## Experiment study

In this section, we describe the datasets used in our experiments. Subsequently, we present a detailed experimental methodology for assessing the effectiveness of the proposed approach. Finally, we conducted a comparative analysis of the baseline methods.

## Dataset

To verify the validity of our proposed model, we conducted experiments on the most widely used datasets in NLU research, namely, ATIS (*Hemphill, Godfrey & Doddington, 1990*) and SNIPS (*Coucke et al., 2018*). These datasets were selected in this study for their complementary characteristics: ATIS provides a focused, domain-specific challenge,

**Table 1 Description of ATIS and SNIPS datasets.**

| Dataset | Characteristics | Rationale for Selection |
|---|---|---|
| ATIS | **Domain-Specific:** Focused on air travel information<br>**Imbalanced intent Types:** 75% of the intents belongs to one class (atis_flight)<br>**Historical Significance:** One of the earliest and widely used NLU datasets.<br>**No. of Intents:** 21<br>**No. of slots:** 128<br>**Training/Test/Validation data:** 4478/893/500 | • Well-defined structure for benchmarking.<br>• Facilitates comparison with existing models<br>• Standard benchmark for evaluating advancements in NLU. |
| SNIPS | **Diverse Domains:** Covers multiple domains like music, weather, restaurant bookings, creative works, book rating etc.<br>**Balanced Intent Types:** Balanced intent distribution across different domains<br>**Varied Language Styles:** Includes colloquial language and diverse query structures<br>**Modern Benchmark:** Reflects contemporary conversational patterns and intent types.<br>**No. of Intents:** 7<br>**No. of slots:** 72<br>**Training/Test/Validation data:** 13084/700/700 | • Tests model's generalizability across different domains.<br>• Challenges models with real-world user inputs.<br>• Relevant for evaluating state-of-the-art NLU models |

**Table 2 An example of a semantic frame.**

| Entity | slots | Intent |
|---|---|---|
| I | O | |
| want | O | |
| to | O | |
| fly | O | |
| from | O | |
| Baltimore | B-fromloc.city_name | atis_flight |
| to | O | |
| Dallas | B-toloc.city_name | |
| round | B-round_trip | |
| trip | I-round_trip | |

whereas SNIPS offers a broader and more diverse set of challenges. Together, they provide a comprehensive test bed for evaluating the effectiveness and generalizability of our proposed model in comparison with the existing models. Table 1 presents the key characteristics of the ATIS and SNIPS datasets.

Table 2 presents an example of a semantic frame from the ATIS dataset, using the sentence "I want to fly from Baltimore to Dallas round trip." The slots use the IOB (in-out-begin) format for slot tagging. In this example, the intent is to find a flight, with 'Baltimore' tagged as the departure city, 'Dallas' as the arrival city, and 'round trip' indicating the type of trip.

## Experimental set-up

We conducted a grid search to select the hyperparameters that would yield the best performance for our model. The grid search involved systematically varying the hyperparameters and evaluating the performance of the model on a validation set. The following selections were made based on the empirical testing and previous research findings:

### Filter sizes and fetaure maps

We explored a range of filter sizes for capturing varying n-gram features in the input text. Specifically, we tested filter sizes of (1, 2, 3, 4, 5, 6) and determined that a combination of filter sizes (2, 3, 5) provided the best balance between capturing short and long-term dependencies. The number of feature maps was also tuned with values ranging from 64 to 256. A value of 128 was chosen because it provides a good trade-off between capturing sufficient detail and computational efficiency.

### Dropout

To prevent overfitting, we experimented with dropout rates from 0.1 to 0.7. A dropout rate of 0.5 was found to be optimal for the feature maps and after the shared encoder, as it effectively regularized the model without significantly impacting training convergence.

### Hidden units and activation function

The number of hidden units was varied from 100 to 300 in increments of 50. The selection of 200 hidden units was based on their ability to offer sufficient capacity for learning complex patterns without overfitting.

### Regularization

L2 regularization is applied to the weights of the dense layers. We experimented with L2 coefficients ranging from 0.0001 to 0.01, finding that 0.01 provided the best regularization effect, helping to prevent the model from becoming complex.

### Optimizer and learning rate

The Adam optimizer was chosen for its efficiency and adaptive learning rate capabilities. We also experimented with learning rates ranging from 0.0001 to 0.01. The default learning rate provided by Adam (0.001) was found to be optimal in this case.

### Batch size

We tested batch sizes of (16, 32, 64, 128) and determined that a batch size of 32 provided the best balance between computational efficiency and model performance.

Table 3 summarizes the selected hyperparameters used in the study, and their selection was guided by both the grid search results and a literature review.

The following metrics were used to evaluate the performance of the model.

### Accuracy

Accuracy is the most commonly used evaluation metric for intent detection. This metric has been used in several related studies (*Liu & Lane, 2016a*; *Qin et al., 2021*; *Saleem & Kim, 2024*). It was computed as the ratio of the number of correct intent predictions to the

**Table 3  Hyperparameters of the proposed model.**

| Hyperparameter | size |
| --- | --- |
| Filter size | [2,3,5] |
| Filter number | 128 |
| Dropout | 0.5 |
| Hidden units | 200 |
| L2 regularization | 0.01 |
| Learning rate | 0.001 |
| Batch size | 32 |

number of utterances, as shown in Eq. (18).

$$Accuracy = \frac{\text{Number of correct intent prediction}}{\text{Number of utterances}} \quad (18)$$

### F1-score

For the slot filling task, F1-score is the most widely used metric employed in previous studies (*Firdaus et al., 2018*; *Guo et al., 2024*; *Hakkani-Tür et al., 2016*; *Zhu et al., 2024*). It is a measure of the average overlap between the ground truth response and prediction (*Saranya & Amutha, 2024*), and is computed using Eq. (19).

$$F1-score = 2 * \frac{Precision.Recall}{Precision + Recall} \quad (19)$$

Where precision and recall are defined as:

$$Precision = \frac{TruePositives}{TruePositives + FalsePositives} \quad (20)$$

$$Recall = \frac{TruePositives}{TruePositives + FalseNegatives} \quad (21)$$

The experiments were conducted in a Jupyter 6.5.4 IDE environment, installed on a Windows 10 platform with an Intel Core i7 processor, 16.0 GB RAM. Python 3.11.5 was used for the experiments with Keras 2.15.0 and TensorFlow 2.15.0, as the primary frameworks for building and training the neural network models. The Pytorch library was also utilized to handle tensor operations and generate BERT embeddings. Numpy 1.24.3, pandas 2.0.3, NLTK 3.8.1, and Gensim 4.3.0, were used in the experiments.

## Comparative methods

To examine the effectiveness of our proposed model, we initially identified the best-performing settings of our model and subsequently compared them with the following baseline models:

- RNN-LSTM (*Hakkani-Tür et al., 2016*): This model uses a BiRNN with LSTM to perform joint learning classification using lexical features represented by 1-hot encoding.
- BiRNN-attention (*Liu & Lane, 2016a*): This model employs an encoder–decoder architecture with an attention mechanism with random embedding initialization.

- CNN-BiLSTM (*Wang, Tang & He, 2018*): Uses CNN to extract features from word2vec word embeddings and the BiLSTM layer as an encoder and decodes it with an attention-based RNN. This baseline model was used as a representative deep learning-based method that exploits the CNN and BiLSTM models.
- BiGRU/BiLSTM-MLP (*Firdaus et al., 2018*): A multitask ensemble method with features obtained from glove, word2vec, and POS tags.
- BiLSTM-self-attention (*Qin et al., 2019*): This model uses stack propagation, which utilizes intent information to guide the slot filling task.
- BiLSTM+Attention (*Chao, Ke & Xiaofei, 2020*): A joint model with POS scaling attention to help the model focus on verbs and nouns that are important in representing user behavior and object operations, respectively.
- SASGBC (BERT only) (*Wang, Huang & Hu, 2020*): This model uses BERT to encode an input sequence, integrate intent information with slot gates, and establish a contextual semantic relationship with self-attention.
- BiLSTM+attention (*Qin et al., 2021*): This model exploits the pretrained BERT model together with BiLSTM and co-interactive attention, and initializes the embedding layer with glove embeddings.
- BERT (*Hardalov, Koychev & Nakov, 2023*): A contextual BERT model is used to design a joint model with name entity recognition(NER) features.

It should be noted that following the common practice in the literature (*Firdaus et al., 2018*; *He et al., 2021*), this study directly used the findings of the aforementioned state-of-the-art methods presented in their original publications and compared them with our proposed model.

## EXPERIMENTAL RESULT AND ANALYSIS

Table 4 and Fig. 2 illustrate the  performance of different variations of the proposed model. MCNN-BiLSTM-1 yielded inferior results compared to MCNN-BiLSTM-2a, MCNN-BiLSTM-2b, and MCNN-BiLSTM-3, which benefitted from the pretrained embeddings. This highlights the effectiveness of pretrained word embedding over randomly initialized embedding in the joint learning classification of intent detection and slot filling, which is consistent with prior research findings (*Bhasin et al., 2020*; *Do & Gaspers, 2019*; *Firdaus et al., 2021*). Similarly, one can notice that MCNN-BiLSTM-2a utilizing general embedding outperforms MCNN-BiLSTM-2b with domain-specific embedding. This is likely because general embeddings capture a broader range of linguistic patterns and semantic relationships than domain-specific ones, which may not adequately represent the complexities of the target domain. Additionally, MCNN-BiLSTM-4 outperformed the versions with single embedding, emphasizing the benefits of incorporating different embeddings and syntactic information into the joint model, which aligns with previous research findings (*Firdaus et al., 2018*; *Siddhant, Goyal & Metallinou, 2019*). Moreover, the MCNN-BiLSTM-5 results show the benefit of harnessing POS tag embedding for better sequence labeling. This can be attributed to the POS tag helping to disambiguate

**Table 4  Performance of different versions of the proposed model.**

| Model | Embeddings | ATIS | | SNIPS | |
|-------|-----------|------|------|-------|------|
| | | Intent (Accuracy) | Slot (F1-score) | Intent (Accuracy) | Slot (F1-score) |
| MCNN-BiLSTM-1 | Random | 79.00 | 96.04 | 86.79 | 94.01 |
| MCNN-BiLSTM-2a | Word2vec | 94.27 | 97.93 | 95.04 | 95.45 |
| MCNN-BiLSTM-2b | Domain-specific word2vec | 93.48 | 96.12 | 88.10 | 95.04 |
| MCNN-BiLSTM-3 | BERT | 96.25 | 96.02 | 97.24 | 94.46 |
| MCNN-BiLSTM-4 | BERT + wor2vec | 97.28 | 98.41 | 97.45 | 96.96 |
| MCNN-BiLSTM-5 | Word2vec + POS | 94.80 | 98.62 | 97.58 | 96.52 |
| MCNN-BiLSTM-6 | BERT + POS | 96.31 | 94.83 | 98.07 | 94.38 |
| MCNN-BiLSTM-7a | BERT + domain-specific word2vec + POS | 96.75 | 98.72 | 97.89 | 97.04 |
| MCNN-BiLSTM-7b | BERT + word2vec + POS | **97.90** | **98.86** | **98.88** | **97.07** |

**Notes.**
The best results are shown in bold.

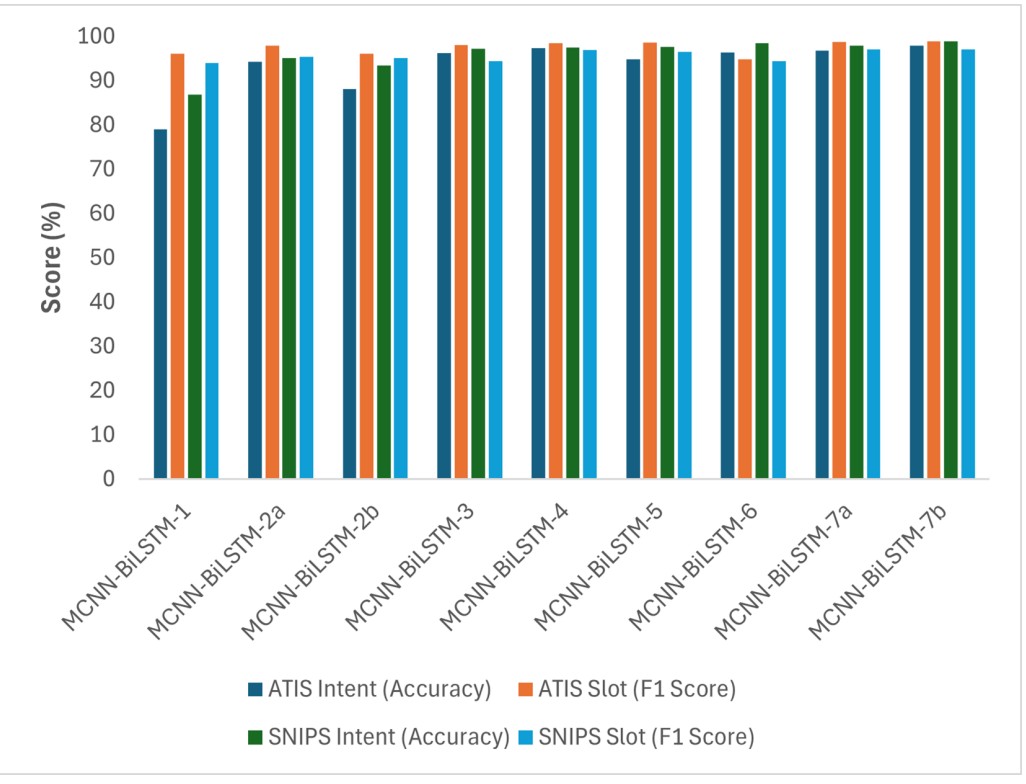

**Figure 2  Graphical representation of the performances of the different versions of the proposed model based on accuracy and F1-score.**

non-contextual embedding, although it has a lesser effect on contextual embedding, as seen in MCNN-BiLSTM-6.

The results clearly indicate that MCNN-BiLSTM-7b outperformed all the other model versions across all datasets. These findings emphasize the advantages of utilizing three

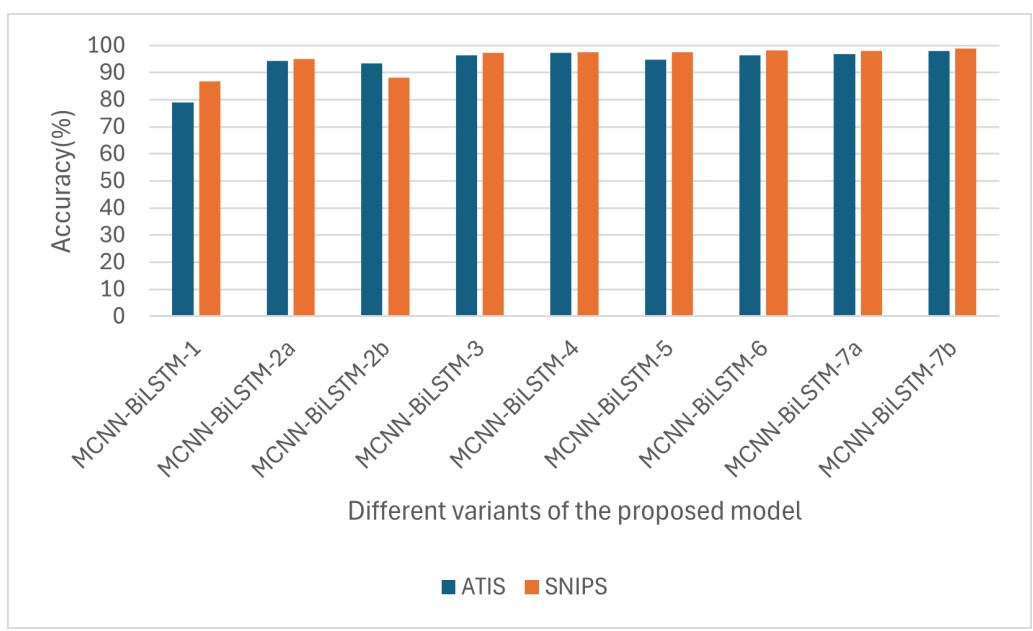

**Figure 3** Performances of the different versions of the proposed model based on accuracy.

input vectors to enhance the accuracy and F1-score. This is likely owing to the context-rich embeddings provided by BERT, which are crucial for intent detection, whereas word2vec embeddings aid in understanding the semantic relationship between slots. However, it is worth noting that the distribution hypothesis problem influences word2vec embedding. Therefore, the inclusion of POS tag embedding, which is an important linguistic feature, helps mitigate this issue by disambiguating words and ultimately improving model performance. The MCNN-BiLSTM-7a results also show that using non-contextual general word embedding for the ATIS and SNIPS datasets performed better than domain-specific non-contextual word embedding. This finding supports the idea that domain-specific embedding of small datasets results in noisy outputs (*Sarma, Liang & Sethares, 2018*).

Similarly, as shown in Fig. 3, in all the variants, the intent accuracy for SNIPS is always higher than that of ATIS, which is likely due to the balanced nature of the SNIPS dataset intents compared to that of the ATIS datasets, which has 75% of its datasets belonging to a single intent.

For the F1-score, one can notice that the F1-score for ATIS is higher than that of SNIPS, as shown in Fig. 4. This is also likely due to the sequence length; the maximum sequence length for the ATIS dataset is higher than that for the SNIPS dataset, and RNN-based models have difficulty capturing long-range dependencies.

## COMPARISON WITH EXISTING METHODS

To better evaluate the effectiveness of our proposed approach, we conducted a comparison with the existing methods. Specifically, we selected the best-performing setting of our proposed model, MCNN-BiLSTM-7b, as presented in Table 4 and Fig. 2, and compared it

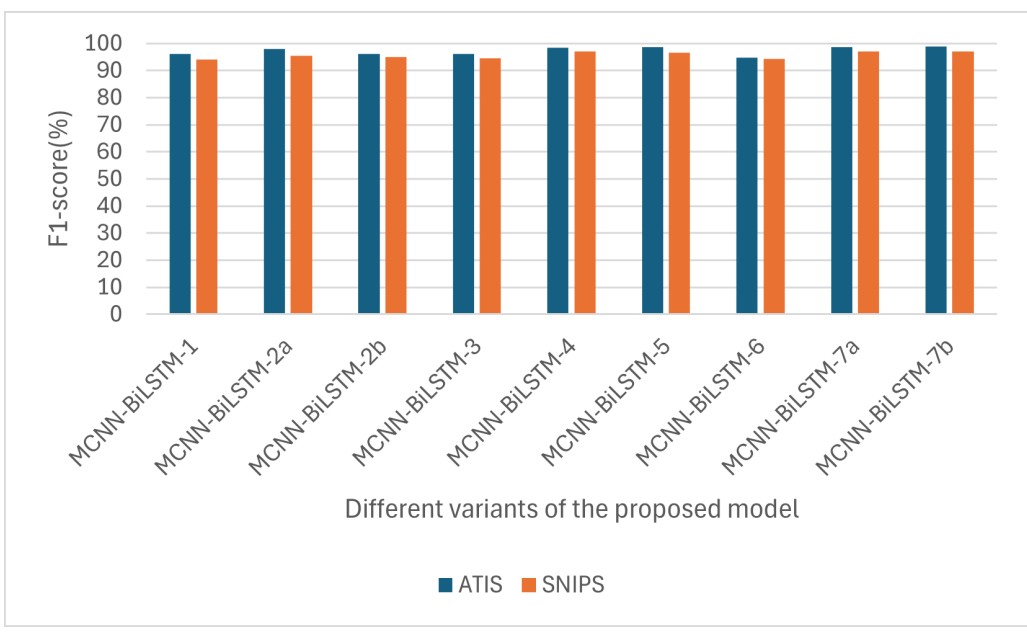

**Figure 4   Performances of the different versions of the proposed model based on F1-score.**

**Table 5   Results of the proposed method compared with the baseline methods.**

| Model | Features | ATIS Dataset | | SNIPS | |
|---|---|---|---|---|---|
| | | Accuracy | F1-score | Accuracy | F1-score |
| RNN-LSTM (*Hakkani-Tür et al., 2016*) | Lexical features (one-hot encoding) | 94.6 | 89.40 | – | – |
| BiRNN + Attention(*Liu & Lane, 2016a*) | Random embeddings | 94.4 | 95.78 | – | – |
| CNN-BiLSTM (*Wang, Tang & He, 2018*) | Word embeddings(word2vec) | 97.17 | 97.76 | – | – |
| Bi-GRU + feature (*Firdaus et al., 2018*) | Glove + Word2vec + POS | 97.76 | 97.93 | – | – |
| BiLSTM+Attention (*Chao, Ke & Xiaofei, 2020*) | POS | 95.70 | 95.60 | 97.70 | 89.2 |
| BC (*Wang, Huang & Hu, 2020*) | BERT embeddings | 97.20 | 96.34 | 98.0 | 95.68 |
| BiLSTM+BERT (*Qin et al., 2021*) | Word embeddings (Glove) | 97.70 | 95.5 | 98.80 | 95.9 |
| Enriched Transformer (*Hardalov, Koychev & Nakov, 2023*) | BERT embeddings + NER | 97.87 | 96.25 | 98.86 | 96.57 |
| **MCNN-BiLSTM-7b** | **Word2vec + BERT + POS** | **97.90** | **98.86** | **98.88** | **97.07** |

**Notes.**
The best results are shown in bold.

with other existing methods. Table 5 and Fig. 5 present the performance of our proposed approach in comparison with the state-of-the-art methods on the ATIS and SNIPS datasets. The best performing variant of the proposed model outperformed the state-of-the-art approaches.

Compared to the RNN-LSTM model, which relies on lexical features encoded using the classical one-hot encoding model, our model demonstrated superior performance. Specifically, our model shows an improvement of 3.3% in accuracy and 3.26% in the F1-score on the ATIS dataset. This underscores the advantage of using word embeddings in deep learning models instead of classical embeddings such as one-hot encoding.

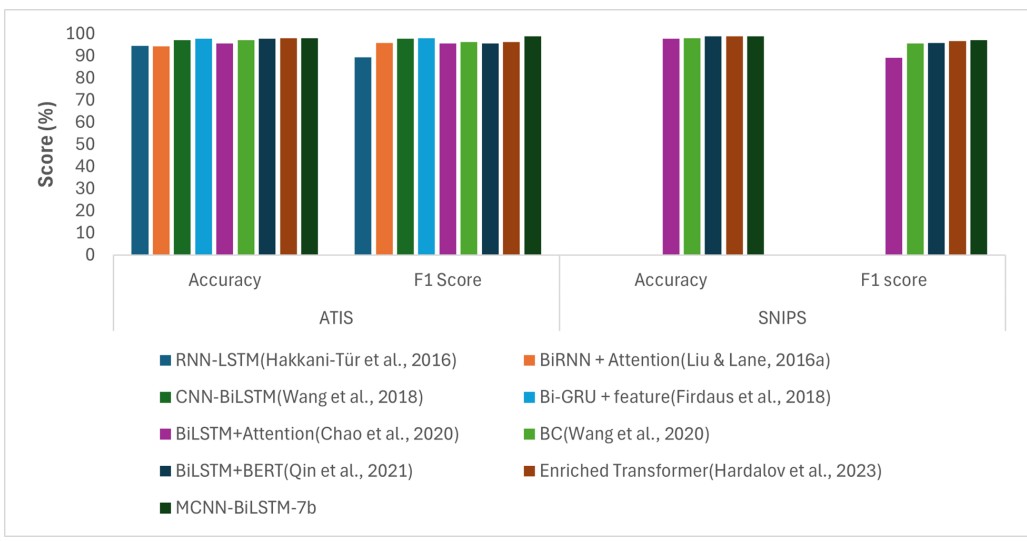

**Figure 5** Graphical representation of the results of the proposed model compared to baselines.

Furthermore, the proposed model outperformed the BiRNN-attention model, which utilized random embeddings with notable gains of 3.5% and 3.08% in accuracy and F1-score, respectively. This could be due to the random initialization of the embeddings, which were assigned arbitrarily, without considering the semantic meanings of the input words.

When compared to CNN-BiLSTM model, which is based on CNN and BiLSTM and utilizes word2vec embeddings, our proposed method outperforms it with gains of 0.73% accuracy and 1.1% F1-score. This is because CNN-BiLSTM utilizes only one input layer compared to our proposed model, which utilizes three input vectors: non-contextual embeddings (word2vec), contextual embeddings (BERT), and syntactic embeddings (POS), as opposed to reliance on single embeddings. This leverages the advantages of the proposed model over traditional CNN-based models in leveraging multiple input channels for improved performance. Compared with BiGRU, which utilizes two non-contextual embeddings with POS features, our proposed model outperformed it with gains of 0.14% and 0.93% in accuracy and F1-score, respectively, on the ATIS dataset. The advantage of our proposed model is likely to be due to the use of contextual embedding. Similarly, compared with the BiLSTM-attention model that uses POS features, our model outperformed it on both datasets with gains of 2.2% and 1.18% accuracy and 1.18% and 7.87% F1-score, respectively. This shows that using external features such as POS and NER alone might not allow a model to attain its ultimate performance. However, they should also be used as supplementary features for other word embedding. When compared with the BC model that utilizes BERT embeddings, our proposed model outperformed it with gains of 0.7% increase in accuracy and 2.52% increase in F1-score for the ATIS dataset, and 0.88% increase in accuracy and 1.39% increase in F1-score for the SNIPS dataset. Table 5 also shows that our model outperformed the BiLSTM-BERT model that utilized word2vec embeddings with a gain of 0.2% increase in accuracy and 3.36% increase in F1-score for

**Table 6** *T*-test between the proposed model and the baselines on ATIS dataset in terms of accuracy.

| MCNN-BiLSTM-6 Vs Baselines | | Paired Difference | | | | | | t | Sig. (2-tailed) |
|---|---|---|---|---|---|---|---|---|---|
| | | Mean | Std. Deviation | Std. Error Mean | 95% Confidence Interval of the difference | | | | |
| | | | | | Lower | Upper | | | |
| Pair 1 | 6-RNN | 3.30000 | 0.02236 | 0.01000 | 3.27224 | 3.32776 | | 330.000 | 5.059E−10 |
| Pair 2 | 6-BiRNN | 3.50000 | 0.02236 | 0.01000 | 3.47224 | 3.52776 | | 350.000 | 3.9981E−10 |
| Pair 3 | 6-CNN | 0.73000 | 0.02236 | 0.01000 | 0.70224 | 0.75776 | | 73.0000 | 2.1102E−7 |
| Pair 4 | 6-BiGRU | 0.140000 | 0.02236 | 0.01000 | 0.11224 | 0.16776 | | 14.0000 | 1.5101E−4 |
| Pair5 | 6-BiLSTM | 2.20000 | 0.02236 | 0.01000 | 2.17224 | 2.22776 | | 220.000 | 2.5609E−9 |
| Pair6 | 6-BC | 0.70000 | 0.02236 | 0.01000 | 0.67224 | 0.72776 | | 70.000 | 2.4956E−7 |
| Pair7 | 6-BiLSTM+BERT | 0.20000 | 0.02236 | 0.01000 | 0.17224 | 0.22776 | | 20.000 | 3.6883E−5 |
| Pair8 | 6-Transfomer | 0.03000 | 0.02236 | 0.01000 | 0.00224 | 0.05776 | | 3.000 | 0.04000000 |

the ATIS dataset and 0.08% increase in accuracy and 1.17% increase in F1-score for the SNIPS dataset. Moreover, despite the use of NER features and BERT embeddings in an enriched transformer model, our proposed model outperformed it with a 0.03% gain in accuracy and 2.61% gain in F1-score for the ATIS dataset and 0.02% gain in accuracy and 0.5% gain in F1-score for the SNIPS datasets. This demonstrates the benefit of harnessing the additional features of non-contextual embeddings in our proposed model.

The key factor responsible for the good performance of our model compared to the baseline models is the use of multiple embeddings obtained through a multichannel convolutional neural network, which has been proven to perform well on some tasks in the NLP field. The POS tag embedding layer disambiguates word2vec embeddings for better sequence labeling. In contrast, the BERT embedding layer input improved intent detection. Another key factor is the use of the BiLSTM layer over the MCNN layer to obtain a contextual utterance representation for the intent and slot detection. This model is relatively uncomplicated and requires no feature engineering, which saves time and improves performance.

## Statistical analysis of the proposed model performance in comparison to the baseline

To validate the significance of the improvements in our proposed model compared to state-of-the-art baseline methods, we used a statistical package for social sciences to conduct a statistical significance assessment using a *t*-test with a significance level of 0.05. A *t*-test yielding a *p*-value less than 0.05 indicates that the observed differences are statistically significant, suggesting less than 5% probability that these differences occurred by chance, thereby confirming a substantial difference between the outcomes of the two groups.

Tables 6 and 7 present the results of our statistical analyses on the accuracy and F1-score metrics, respectively, based on a five-fold validation on the ATIS dataset. Notably, the computed *p*-values in these tables were below the critical significance level of 0.05,

**Table 7** *T*-test between the proposed model and the baselines on ATIS dataset in terms of F1-score.

| MCNN-BiLSTM-6 Vs Baselines | | Paired Difference | | | | | t | Sig. (2-tailed) |
|---|---|---|---|---|---|---|---|---|
| | | Mean | Std. deviation | Std. Error Mean | 95% Confidence Interval of the difference | | | |
| | | | | | Lower | Upper | | |
| Pair 1 | 6-RNN | 9.46000 | 0.03162 | 0.01414 | 9.42074 | 9.49926 | 668.923 | 2.4951E−7 |
| Pair 2 | 6-BiRNN | 3.08000 | 0.03162 | 0.01414 | 3.04074 | 3.11926 | 217.89 | 2.6665E−9 |
| Pair 3 | 6-CNN | 1.10000 | 0.03162 | 0.01414 | 1.06074 | 1.13926 | 77.782 | 1.6374E−7 |
| Pair 4 | 6-BiGRU | 0.93000 | 0.03162 | 0.01414 | 0.89074 | 0.96926 | 65.761 | 3.2034E−7 |
| Pair5 | 6-BiLSTM | 3.26000 | 0.03162 | 0.01414 | 3.22074 | 3.29926 | 230.517 | 2.1246E−9 |
| Pair6 | 6-BC | 2.52000 | 0.03162 | 0.01414 | 2.48074 | 2.55926 | 178.191 | 5.9500E−9 |
| Pair7 | 6-BiLSTM+BERT | 3.36000 | 0.03162 | 0.01414 | 3.32074 | 3.39926 | 237.588 | 1.8828E−9 |
| Pair8 | 6-Transfomer | 2.61000 | 0.03162 | 0.01414 | 2.57074 | 2.64926 | 184.555 | 5.1709E−9 |

**Table 8** *T*-test between the proposed model and the baselines on SNIPS dataset in terms of accuracy.

| MCNN-BiLSTM-6 Vs Baselines | | Paired Difference | | | | | t | Sig. (2-tailed) |
|---|---|---|---|---|---|---|---|---|
| | | Mean | Std. deviation | Std. Error Mean | 95% Confidence Interval of the difference | | | |
| | | | | | Lower | Upper | | |
| Pair1 | 6-BiLSTM | 1.18000 | 0.05831 | 0.02608 | 1.10760 | 1.25240 | 45.251 | 1.4264E−6 |
| Pair2 | 6-BC | 0.88000 | 0.05831 | 0.02608 | 0.80760 | 0.95240 | 33.746 | 4.5994E−6 |
| Pair3 | 6-BiLSTM+BERT | 0.08000 | 0.05831 | 0.02608 | 0.00760 | 0.15240 | 3.068 | 0.03700 |
| Pair4 | 6-Transfomer | 0.02000 | 0.05831 | 0.02608 | −0.05240 | 0.09240 | 0.767 | 0.486000 |

**Table 9** *T*-test between the proposed model and the baselines on SNIPS dataset in terms of F1-score.

| MCNN-BiLSTM-6 Vs Baselines | | Paired Difference | | | | | t | Sig. (2-tailed) |
|---|---|---|---|---|---|---|---|---|
| | | Mean | Std. deviation | Std. Error Mean | 95% Confidence Interval of the Difference | | | |
| | | | | | Lower | Upper | | |
| Pair1 | 6-BiLSTM | 7.87000 | 0.10440 | 0.04669 | 7.74037 | 7.99963 | 168.557 | 3.1597E−10 |
| Pair2 | 6-BC | 1.39000 | 0.10440 | 0.04669 | 1.26037 | 1.51963 | 29.771 | 2.6447E−8 |
| Pair3 | 6-BiLSTM+BERT | 1.17000 | 0.10440 | 0.04669 | 1.04037 | 1.29963 | 25.059 | 3.5163E−8 |
| Pair4 | 6-Transfomer | 0.50000 | 0.10440 | 0.04669 | 0.37037 | 0.62963 | 10.709 | 9.7354E−8 |

indicating statistical significance. The statistical analyses of accuracy and F1-score metrics are presented in Tables 8 and 9, respectively. The results show that the performance of the proposed model is statistically significant based on the F1-scores (Table 9). However, in Table 8, the baseline model proposed by *Hardalov, Koychev & Nakov (2023)*, which uses an enriched BERT model, shows no statistically significant difference from the proposed model. This outcome can be attributed to the fine-tuning of the entire BERT transformer, capturing a richer task-specific context. However, it is computationally more expensive compared to using static BERT embeddings, as employed in our proposed model.

## CONCLUSION AND FUTURE WORK

The motivation for this study arises from the need to enhance dialogue systems' performance in real-world scenarios, where accurate and nuanced language understanding is crucial. Current models often struggle with complex language structures, particularly in tasks like intent detection and slot filling, due to a lack of comprehensive integration of contextual, syntactic, and semantic features. To address these challenges, this study focused on developing a joint model that integrates these diverse linguistic features to improve performance. The experimental results demonstrated that a multichannel convolutional neural network architecture with three distinct channels for non-contextual, contextual, and part-of-speech tag embeddings significantly enhanced both intent detection and slot filling tasks. Specifically, the use of contextual embeddings improved the model's ability to understand complex language structures, which is vital for accurately capturing user intent. Non-contextual embeddings, which achieved higher F1-scores, proved more effective for slot filling tasks, while general embeddings showed broader applicability than domain-specific ones, especially for smaller datasets. This highlights the importance of embedding diversity in enhancing model performance. Additionally, the integration of part-of-speech tag embeddings contributed to reducing errors in slot filling by disambiguating polysemous words. Building on the strengths of our approach, future research could explore methods to dynamically integrate and weigh different types of embeddings based on the context and specific characteristics of the dataset. Beyond part-of-speech tags, the inclusion of other linguistic features, such as named entity recognition tags or dependency parsing information, could further enhance the model's understanding. Furthermore, addressing dataset imbalances, like those present in ATIS, through data augmentation techniques or synthetic data generation, offers another avenue for improvement.

### Funding

The research was funded by the Ministry of Higher Education Malaysia and the Universiti Teknologi Malaysia (UTM) under grant scheme FRGS/1/2022/ICT06/UTM/01/1 with grant vote number R.J130000.7851.5F568. There was no additional external funding received for this study. The funders had no role in study design, data collection and analysis, decision to publish, or preparation of the manuscript.

## Grant Disclosures

The following grant information was disclosed by the authors:

Ministry of Higher Education Malaysia.

Universiti Teknologi Malaysia(UTM): FRGS/1/2022/ICT06/UTM/01/1, R.J130000.7851.5F568.

## Competing Interests

The authors declare there are no competing interests.

## Author Contributions

- Yusuf Idris Muhammad conceived and designed the experiments, performed the experiments, performed the computation work, prepared figures and/or tables, authored or reviewed drafts of the article, and approved the final draft.
- Naomie Salim analyzed the data, authored or reviewed drafts of the article, and approved the final draft.
- Anazida Zainal analyzed the data, authored or reviewed drafts of the article, and approved the final draft.

## Data Availability

The ATIS and SNIPS training, test and dev. samples, and the code for data preparation, the embedding code and train and evaluation code, and the domain adapted embedding code are all available in the Supplemental Files.

## Supplemental Information

Supplemental information for this article can be found online at http://dx.doi.org/10.7717/peerj-cs.2346#supplemental-information.

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
