# Peer review of "Joint intent detection and slot filling with syntactic and semantic features using multichannel CNN-BiLSTM"

_PeerJ Computer Science, doi:10.7717/peerj-cs.2346_

## Round 0.1 · original submission · Major Revisions

Dear authors,

Thank you for submitting your article. Feedback from the reviewers is now available. It is not recommended that your article be published in its current format. However, we strongly recommend that you address the issues raised by the reviewers, especially those related to readability, experimental design and validity, and resubmit your paper after making the necessary changes.

Best wishes,

Reviewer 1 ·

Basic reporting

First of all, I would like to congratulate the author(s) for their work.

Please be assured that I will evaluate the study only from a scientific point of view, without prejudice.

This work is valuable and prudent work in the computer science community and social media area.

Also, I can't end without saying that it is worth reading.

In the study, the author(s) evaluated by using multichannel CNN and BiLSTM approach models to predict intent detection and slot filling.

Also, this paper illustrated the potential for intent detection and slot filling to enhance recommendation systems by providing a more comprehensive understanding of users’ preferences and overcoming the limitations of traditional rating-based approaches.

Finally, the authors claim that the proposed multichannel CNN and BiLSTM models perform better than the other classical methods.

The results obtained are presented in detail with tables and graphs.

Experimental design

The article should be prepared by considering the journal template writing rules (see: https://peerj.com/about/author-instructions/cs).

Although the article was prepared with great effort, there are typographical errors in the most prominent parts of the article.

These deficiencies can't be tolerated for the valuable study.

- From the use of abbreviations to reference notation, the article should be completely revised.

ex1. Non-contextual word embeddings capture static meanings but lack contextual information, while contextual embeddings like BERT provide deeper understanding by considering surrounding words, though they struggle with out-of-vocabulary (OOV) words. This study proposes a hybridized text representation method for a joint model using a multichannel CNN.

ex2. These embeddings were fed into a convolutional layer, concatenated into a one-dimensional vector, pooled, and passed through a shared BiLSTM to obtain a deep contextual representation of the input sequence.

- Also, choose correctly the verbs that emphasize the achievement results of the study.

ex3: Experiments using the ATIS and SNIPS datasets showed that this approach outperforms baseline models, demonstrating the effectiveness of the proposed method in improving performance on both tasks. Additionally, there is potential to investigate the development and use of instruments specific to the linguistic and cultural characteristics of the environment being studied.

- Attention should be paid to the use of abbreviations.

ex4: The goal of NLU is to extract meaning from a users utterances and to infer their intentions. NLU has two primary tasks: Intent Detection (ID) and Slot Filling (SF). ID is a classification problem, whereas SF is a sequence labeling problem. Previous studies have traditionally treated intent detection and slot filling as separate tasks.

ex5: Deep neural networks have revolutionized natural language processing tasks such as intent detection and slot filling. Commonly used intent detection models include convolutional neural networks (CNNs), recurrent neural networks (RNNs), and transformer models such as bidirectional encoder representations from transformers (BERT)(Khattak et al., 2021).

- Some of your sentences should be shortened and clarified.

- Incomplete and inconsistent sentences should be completed. In addition, the article should be cleared of all grammatical errors by using the grammar check tool (For example, Grammarly or Ginger).

Validity of the findings

In the abstract section, clearly emphasize your main motivation for the preparation of the article.

Also, express numerically all the performance metrics obtained by the method proposed in the summary section.

The last sentence of the Abstract section must have been completed with a more striking sentence.

In the conclusion section, list your achievements with this study.

Clearly express your contribution to future studies and literature.

Compare your performance metrics with previous studies in the literature.

Satisfactory technical information about the method used is not presented.

No mathematical expressions, from the evaluation metrics used to the algorithms, are included in the article.

Please clearly describe the methods and algorithms used.

Specify the parameters of the algorithms used separately.

If possible, add innovative classifiers such as deep learning or optimization algorithms. It is left to the discretion of the author(s).

Provide the motivation for the conclusion part.

If possible, include a flowchart that will summarize the study.

Reviewer 2 ·

Basic reporting

No comment

Experimental design

The authors are required to provide more details about the datasets used (e.g., specific characteristics, why they were chosen). This information can be included in a table format for clarity.

While hyperparameters are mentioned, it would be helpful to explain the rationale behind their selection. Also, discuss any hyperparameter tuning process.

Validity of the findings

The authors should provide more detail on the experimental setup, such as the environment in which experiments were run, the specifications of the hardware used, and any specific software or libraries utilized.

They should also include statistical tests (e.g., t-tests, p-values) to show the significance of the improvements over baseline models. Discuss the confidence intervals for the reported metrics.

Additional comments

• The title conveys what the text is all about, although it could still be a little more concise. The title should be shortened while structural elements are maintained.
• The abstract should be more precise and concise by focusing on the primary contributions and results. For example, the discussion on non-contextual and contextual embeddings can be shortened.
• Although the introduction provides a good background of the study, but it lacks a strong motivation to the study. Try to include a paragraph on the importance and potential applications of improved intent detection and slot filling in real-world scenarios.
• Even though relevant literatures are cited, the related work section could benefit from a brief critique of the studies reviewed to highlight gaps that your work aims to address.
• The related work sections could be more structured by organizing it into sub-sections. For instance, classical approaches, deep learning approaches, hybrid models etc
• Discuss how your approach differs and potentially improves upon these methods
• The authors should consider a detailed diagram of the proposed model architecture as this will aid the understanding of the proposed study and ensure all components of the model are clearly labelled and described in the caption.
• Provide more details about the dataset used such as specific characteristics, why you chose them. This information can be included in a table format for clarity.
• In the conclusion, the authors should emphasize the significance of the improvement achieved and summarizes the key findings. The future works should be more specific and suggest area of exploration based on the study findings.
• The technical details of the model are too complex, consider simplifying the language.
• The article lacks citations from recent publications, specifically from this year 2024.

---

## Round 0.2 · accepted · Accept

Dear authors,

Thank you for the revision and for clearly addressing all the reviewers' comments. I confirm that the paper has been improved. With this revision, your paper is now acceptable for publication.

Best wishes,

Reviewer 1 ·

Basic reporting

I congratulate the author(s) who contributed to the preparation of the article for their efforts.

The article has been prepared taking into account the journal template writing rules.

Typographical errors have been corrected in important parts of the article.

The article has been completely revised, from the use of abbreviations to references.

Sentences have been shortened and clarified throughout the article.

Incomplete and inconsistent sentences have been completed. In addition, the article has been cleaned of all grammatical errors using the grammar checker.

Experimental design

In the abstract section, your main motivation for the preparation of the article is clearly emphasized.

In addition, all performance metrics obtained with the proposed method are expressed numerically in the abstract section.

The last sentence of the abstract section is completed with a more striking sentence.

In the conclusion section, the achievements achieved with this study are listed.

Validity of the findings

Your contribution to future studies and literature is clearly stated.

Satisfactory technical information about the method used is provided.

The methods and algorithms used are clearly defined.

The parameters of the algorithms used are specified separately.

The motivation for the conclusion section is provided.

A flow chart is added to summarize the study.

Additional comments

All requested changes have been completed in this revision study.

I congratulate the author(s) for their meticulousness.

Reviewer 2 ·

Basic reporting

Thank you for your thorough and diligent revisions in response to the reviewers' comments. After carefully reviewing the revised manuscript, I am pleased to inform you that the paper has been accepted for publication.

The revisions have addressed all the concerns raised by the reviewers, and the manuscript is now significantly improved. Your efforts in enhancing the clarity, depth, and overall quality of the work are commendable.

Congratulations on the acceptance of your paper, and we look forward to its contribution to the field.

Experimental design

No commentN

Validity of the findings

No comment

Additional comments

No comment